# Current and Future Therapeutics for Treating Patients with Sickle Cell Disease

**DOI:** 10.3390/cells13100848

**Published:** 2024-05-16

**Authors:** Mariam Barak, Christopher Hu, Alicia Matthews, Yolanda M. Fortenberry

**Affiliations:** Biology Department, Case Western Reserve University, Cleveland, OH 44106, USA; mariam.barak@case.edu (M.B.); cxh633@case.edu (C.H.); axm1851@case.edu (A.M.)

**Keywords:** sickle cell disease, hydroxyurea, vaso-occlusion, voxelotor, therapeutics, gene therapy, clinical trials

## Abstract

Sickle cell disease (SCD) is the most common genetic blood disorder in the United States, with over 100,000 people suffering from this debilitating disease. SCD is caused by abnormal hemoglobin (Hb) variants that interfere with normal red blood cell (RBC) function. Research on SCD has led to the development and approval of several new SCD therapies in recent years. The recent FDA-approved novel gene therapies are potentially curative, giving patients an additional option besides a hematopoietic bone marrow transplant. Despite the promise of existing therapies, questions remain regarding their long-term pharmacological effects on adults and children. These questions, along with the exorbitant cost of the new gene therapies, justify additional research into more effective therapeutic options. Continual research in this field focuses on not only developing cheaper, more effective cures/treatments but also investigating the physiological effects of the current therapies on SCD patients, particularly on the brain and kidneys. In this article, we undertake a comprehensive review of ongoing clinical trials with completion dates in 2024 or later. Our exploration provides insights into the landscape of current therapeutics and emerging novel therapies designed to combat and potentially eradicate SCD, including the latest FDA-approved gene therapies.

## 1. Introduction

Sickle cell disease (SCD) is a debilitating autosomal recessive disorder that impacts an estimated 100,000 individuals in the United States, with approximately 40,000 of them being children [1,2]. Within this population, approximately 90% are of African American descent [3,4], although SCD also affects individuals of Mediterranean, Middle Eastern, and Indian heritage [3]. SCD manifests in three primary forms: sickle cell anemia (SCA; HbSS), sickle hemoglobin-C (HbSC), and sickle cell beta-thalassemia (HbS/β Th). SCA (HbSS) stands as the most prevalent type, constituting 60% to 70% of the affected individuals in the United States and millions more worldwide [5,6].

In individuals with HbSS, hemoglobin-S binds to adjacent hemoglobin-S molecules, resulting in the formation of long, branched polymers [7]. These polymers are responsible for the characteristic crescent shape of red blood cells in SCD. Furthermore, the red blood cells (RBCs) of individuals with HbSS exhibit elevated levels of reactive oxygen species (ROS), reduced solubility, impaired oxygen transport, irregular quaternary structure, and protein precipitation, leading to damage to the cytoskeleton [8,9]. Consequently, in conjunction with hemoglobin polymerization, the red blood cells undergo irreversible sickling and experience a shortened lifespan of approximately 20 days, compared to the normal lifespan of 120 days for healthy red blood cells [10]. Complications observed in patients with SCD include vaso-occlusive crisis (VOC), acute chest syndrome, hemolytic anemia, and multi-organ failure resulting from capillary flow obstructions to vital organs (Figure 1).

The primary cause of hospitalization in SCD patients is the occlusion of blood vessels resulting in ischemic pain [11]. The frequency of VOCs and other comorbidities, such as acute chest syndrome, can serve as predictors of mortality in individuals with SCD [12]. VOCs occur due to the damage inflicted by sickled RBCs on small blood vessels, leading to inflammation of the endothelial layer and the subsequent activation of platelets, neutrophils, leukocytes, monocytes, and mast cells [12,13,14,15,16]. Additionally, the contents of damaged RBCs activate the inflammatory response and cytokines such as interleukin-1β (IL-1β) and high-sensitivity C-reactive protein (hsCRP), further contributing to VOCs [17,18]. The accumulation of damaged RBCs, inflammatory responses, and the occlusion of small blood vessels ultimately lead to tissue damage, hypoxia, and chronic pain episodes [19].

Several therapeutics are utilized to manage SCD, with hydroxyurea being the most commonly prescribed. Currently, six FDA-approved drugs target acute complications of SCD, including hydroxyurea (approved in 1998), L-glutamine (approved in 2017), crizanlizumab-tmca (approved in 2019), voxelotor (approved in 2019), Casgevy/Exa-cel (approved in 2023), and Lyfgenia/Lov-cel (approved in 2023) (Figure 2). Hematopoietic stem cell transplant (HSCT) stands as the sole curative option for SCD, with treatment success contingent upon factors such as donor compatibility, recipient age, and disease severity [20]. Preferably, stem cells for HSCT are sourced from a matched related donor, as utilizing unmatched donors may lead to complications such as graft-versus-host disease (GVHD) [20]. Addressing HSCT-related complications remains a significant challenge, prompting ongoing research into novel treatment strategies, including NiCord^TM^ [21]. This review delves into ongoing clinical trials, particularly those investigating therapeutics for managing SCD and potential cures for this debilitating condition, including innovative gene therapy approaches.

## 2. Hemoglobin F Induction Therapies

### 2.1. Hydroxyurea (HU)

HU is an antimetabolite that increases fetal hemoglobin (HbF) expression, resulting in a decrease in the frequency and severity of SCD complications. The mechanism by which HU increases the expression of HbF remains under investigation. In 1995, the landmark Multicenter Study of Hydroxyurea in Adult Sickle Cell Anemia (MSH) trial demonstrated HU’s effectiveness in preventing painful crises by increasing HbF levels [22]. A follow-up long-term (17.5 years) observational study of the MSH trial participants showed that hydroxyurea reduced mortality rates compared to patients not taking HU [23]. In 1997, the Pediatric Hydroxyurea in Sickle Cell Anemia (PED HUG) study, a phase I/II interventional trial of hydroxyurea conducted in children with sickle cell anemia, demonstrated hydroxyurea’s safety and effectiveness in increasing HbF levels without affecting growth and development in ages 5 to 15 [24,25]. In 1998, hydroxyurea (HU), became the first FDA-approved medication to treat sickle cell disease, but it was not approved for use in children until 2007 [2] (Figure 2). Ongoing clinical trials on HU focus on its ability to treat other complications, especially ones that affect vital organs such as the brain and kidneys. In this review, we only focus on studies on stroke and the neurological effects of HU on patients with SCD, as HU has been extensively reviewed previously [26].

The BRAINSAFE-II study is an open-label, single-arm clinical trial being conducted in 270 children with sickle cell anemia (ages 3 to 9 years old) in Kampala, Uganda. Patients receive 20 mg/kg of HU per day. The primary outcomes of the study are the frequency, age, and severity of sickle cell vasculopathy manifested by new strokes, TCD velocities, and new or worsening cognitive impairment (NCT04750707). The SACRED study (Stroke Avoidance for Children in Republica Dominicana; NCT02769845), which has been active since March 2016, is examining HU’s effectiveness in reducing stroke in children with sickle cell anemia (aged 3 to 15 years). It is a prospective screening and treatment study with a primary outcome measure of the highest time-averaged maximum velocity (TAMV) obtained in the main intracranial arteries, namely the middle cerebral artery, the internal carotid artery, or the internal carotid bifurcation. HU’s effectiveness will be determined by comparing the baseline and potentially reduced TCD velocities. Preliminary study results from the SACRED study identified a higher prevalence of conditional TCD and lower abnormal TCD, which the investigators suggest could be attributed to differences in baseline status, prior treatment effects, or genetics [27]. The Stroke in Young Adults with Sickle Cell Anemia (SPIYA) study is currently investigating HU’s effectiveness in preventing stroke in Nigerian adolescents and young adults with sickle cell anemia. The prevalence of neurological injury (overt stroke, transient ischemic attacks, and silent cerebral infarcts) during the developmental transition period of age 16 through 25 will be determined. The primary endpoints of the study are the prevalence of neurological morbidity via MRA/MRI, biological morbidity via TCD measurements (>200 cm/s), and risk factors for stroke (NCT04808778). There is one phase IIb, multicenter, double-blinded clinical trial investigating HU’s therapeutic effects on albumin after six months of treatment in adult patients with SCD. The primary endpoint of this study is to detect at least a 30% decrease in the albumin to creatinine ratio (ACR). This study is scheduled to be completed in June 2024 (NCT03806452).

Despite HU’s proven effectiveness for SCD and its potential to treat brain and kidney complications, questions remain concerning its long-term safety. An ongoing open-label extension study called ESCORT-HU (NCT04707235) began in 2020 and is investigating the occurrence of malignancies, leg ulcers, male fertility impairment, and serious unexpected adverse events caused by HU over a 5-year period. These potential adverse effects have also prompted the continued development of alternative HbF induction therapies.

### 2.2. Decitabine

HbF is silenced in infancy by DNA methyltransferase 1 (DNMT1) [28]. Decitabine, in combination with the cytidine deaminase (CDA) inhibitor (tetrahydrouridine (THU)), has been shown to inhibit DNMT1 and induce HbF production [28]. Two trials are recruiting adult SCD patients to investigate decitabine and THU safety and efficacy (NCT04055818 and NCT05405114). Both trials measure changes in hemoglobin levels and have an estimated completion date in late 2024.

### 2.3. HDAC Inhibitors

Previous pre-clinical studies have shown that pan histone deacetylase (pan-HDAC) inhibitors significantly increase HbF induction [29]. A clinical trial on the pan-HDAC inhibitor, panobinostat (LBH589), is recruiting patients to investigate its activity in adult SCD patients (NCT01245179). Other histone deacetylase inhibitors include butyrates (sodium butyrate infusion) [30,31]. Butyrate increases the transcription of the Hb gamma (HBG) gene and the translation of HBG1/HBG2 mRNA [31]. The mean reticulocyte γ-globin chain synthesis increased from 2.6% ± 1.9% to 6.5% ± 2.0% within 24 h of infusion [31]. A phase 2 dose-escalation randomized study examined 2,2-dimethylbutyrate (HQK-1001) with and without hydroxyurea in hemoglobin SS or S/β0 thalassemia patients. They observed a 2% increase in HbF in 21 patients receiving only HQK-1001 and a 2.7% increase in HbF receiving HQK-1001 with hydroxyurea in 31 of their patients [32].

### 2.4. Benserazide

Benserazide is an L-amino acid decarboxylase or DOPA decarboxylase inhibitor that has been used alongside L-DOPA for the treatment of Parkinson’s disease since the 1970s. However, tests have shown that benserazide may also be capable of inducing HbF production through the displacement or suppression of repressors of the γ-globin gene promoter, including BCL11a, LSD-1, KLF-1, and HDAC3 [33]. Clinical trials in anemic baboon models showed that treatment with benserazide at a 3 mg/kg dose twice a week for 2 weeks induced γ-globin mRNA by up to 13-fold, with elevated levels observed for 2 weeks afterward [33]. This makes benserazide a promising alternative for patients who are unresponsive to hydroxyurea treatment. Despite these promising results and in vitro tests suggesting that benserazide could be 20-30 times more potent than hydroxyurea, actual clinical results in 35 patients with Parkinson’s showed that benserazide was not able to appreciably increase HbF production, even at doses 10 times higher than what was expected to produce results [34]. It is noted, however, that none of the patients had SCD, potentially contributing to its lack of effectiveness. There is currently one clinical trial testing benserazide in patients with beta-thalassemia and SCD. The BENeFiTS trial (NCT04432623) began in 2020 and is estimated to be completed in December 2024. The study evaluates three dosages of benserazide, and the primary outcomes are the number of patients with treatment-related adverse events, maximum plasma concentration, and plasma concentration over time. The study also measures the percentage of cells containing HbF, HbF levels, HbF protein per cell, and hemoglobin levels.

## 3. Anti-Polymerization Therapies

### 3.1. Voxelotor (GBT440)

Voxelotor, formerly known as GBT440, inhibits HbS polymerization and RBC sickling by allosterically modifying hemoglobin’s affinity for oxygen [35]. It accomplishes this by reversibly binding to the N-terminal valine of the Hb alpha-chain molecule, stabilizing the RBC in its oxygenated state. As HbS can only polymerize in the deoxygenated state, subsequent sickling and damage to the RBC are reduced [35,36]. This, in turn, improves oxygen delivery to vital organs, and with reduced hemolysis and inflammation, SCD complications such as VOCs and chronic kidney damage (CKD) have been shown to improve with voxelotor [37,38,39,40]. Voxelotor was approved by the FDA in 2019 for the treatment of SCD in both adults and children 12 years and older. In a phase 3 randomized, placebo-controlled trial (Hemoglobin Oxygen Affinity Modulation to Inhibit HbS Polymerization HOPE NCT03036813), completed in 2019, voxelotor successfully inhibited the polymerization of HbS in SCD children and adults (12 years to 65 years). It significantly increased hemoglobin levels and reduced markers of hemolysis [41]. A follow-up longitudinal report of the HOPE patients reported improved hemoglobin levels with the continued use of voxelotor at 1500 mg per day over 72 weeks [42]. Patients who received a higher dosage of voxelotor had higher levels of hemoglobin (51%; 95% confidence interval [CI], 41 to 61) compared to the placebo group (7%; 95% CI, 1 to 12) [41]. The HOPE study concluded that voxelotor is generally safe and well tolerated, with a safety profile similar to a placebo with no substantial related adverse events. Two open-label extension studies are being conducted (NCT03573882 and NCT04188509). The NCT03573882 ongoing study enrolls patients who completed the HOPE study. The study has an enrollment of 435 patients over 100 clinical sites.

The HOPE KIDS-1 study, a phase II, open-label, single- and multiple-dose trial (NCT02850406), examines voxelotor pharmacokinetics, safety, efficacy, and tolerability in pediatric patients aged 6 months to 17 years. Initial findings noted that a small number of patients experienced adverse side effects, the most common being pyrexia, vomiting, rash, abdominal pain, diarrhea, headache, viral infection, pain in extremities, and upper respiratory tract infection [43], which are similar to studies conducted in adults [41]. Voxelotor was otherwise successful in improving hemoglobin levels and markers of hemolysis [43]. HOPE-KIDS 2 (NCT04218084) is an ongoing multicenter, phase 3, randomized, placebo-controlled trial conducted in pediatric SCD patients (aged 2 to 15) with elevated arterial cerebral blood flow. Primary outcomes are changes in TCD flow, and secondary outcomes include changes in unconjugated bilirubin and the annualized incidence of VOCs. The study started in 2020, and its expected completion is in 2025. HEMOPROVE (NCT05199766) is assessing the biological activity of voxelotor in reducing intravascular hemolysis. This study started in 2023 and will be completed in 2025.

In addition to the HOPE studies, a few additional clinical trials are ongoing. Two small clinical studies, NCT05228821 and NCT05018728 (VoxSCAN), with an enrollment of 24 and 50, are focused on assessing voxelotor’s effects on cerebral blood flow. Another small study, with an enrollment of only 10, evaluates voxelotor’s effect on physical performance (NCT06023199). The ReSOLVE study assesses voxelotor’s effect on resolving sickle cell leg ulcers (NCT05561140) in patients 12 years and older. Lastly, a pilot study is currently recruiting participants examining voxelotor’s impact on CKD progression in sickle cell anemia patients with early stages of nephropathy (CKD stage 1 or 2), determined via changes in albuminuria (NCT04335721).

Voxelotor is a new drug that effectively reduces the polymerization of hemoglobin molecules, decreasing sickling. However, since it increases hemoglobin’s affinity for oxygen, there is concern regarding its long-term effects on organ damage due to ischemia, brain function, and tissue oxygenation. Very limited data exist on voxeletor’s effects on VOCs and end-organ damage. Therefore, the results of the ongoing clinical trials and a next-generation voxelotor-type drug aim to address these issues.

### 3.2. GBT021601

GBT021601 is a next-generation voxelotor-type drug that stabilizes Hb in the oxygenated state and inhibits polymerization in SCD subjects [44]. A 2023 study evaluated GBT021601’s effectiveness at reducing the percent of reticulocytes and improving overall survival in Townes sickle cell mice. Townes SS mice were fed either normal chow or chow containing 0.05% or 0.2% GBT021501. Blood was then collected at 1, 14, 31, 43, 55, and 67 weeks and tested for compound exposure and percent of reticulocyte analysis. The study found that GBT021601 achieved dose-dependent reductions in the percent of reticulocytes after just 1 week and increased survival rates in Townes SS mice, achieving a 30% or greater reduction in the percent of reticulocytes [45]. A randomized control study is currently underway to determine the safety, tolerability, efficacy, pharmacokinetics, and pharmacodynamics of GBT021601 in pediatric and adult SCD patients (NCT05431088). A long-term extension study on GBT021601 in SCD patients is also ongoing and measures the long-term effects of the drug on treatment-emergent adverse events, hemolysis, inflammation, quality of life, and reticulocyte levels (NCT05632354). The study enrolled 314 participants and is scheduled to be completed in April 2029.

## 4. Pyruvate Kinase Activators

### 4.1. Mitapivat

Mitapivat (AG-348) is a first-in-class small-molecule allosteric activator of pyruvate kinase and was approved by the FDA in February 2022 for the treatment of hemolytic anemia [46]. However, mitapivat has also shown promise in its ability to treat SCD. Previous research has shown that the activation of erythrocyte pyruvate kinase (PKR) results in the increased production of ATP and the lowered production of 2,3-diphosphoglycerate [47]. This increases Hb’s affinity for oxygen, reducing HbS polymerization as a result. In 2019, a phase 1 trial of mitapivat in 15 SCD patients began and investigated the safety and efficacy of mitapivat as a potential therapy for SCD. Upon the trial’s completion in 2021, the results showed that mitapivat was well tolerated and reduced sickling, improved anemia, reduced markers of hemolysis, and increased oxygen affinity. A European phase 2, open-label, single-center study of mitapivat in patients with SCD (the ESTIMATE study) was completed in 2023 and found that mitapivat was able to significantly reduce the annual rate of VOC, improve levels of hemoglobin, and decrease markers of hemolysis (NL8517). These results support the further evaluation of mitapivat as an SCD therapy, and additional studies are currently ongoing. The RISE UP study began in 2022 and is evaluating the safety and efficacy of mitapivat in 267 patients with SCD (NCT05031780). Annualized rate of pain crises, percentages of patients with Hb response, and percentages of participants with treatment adverse events are being measured, and the study has an estimated completion date of December 2025. Future studies on mitapivat include an upcoming study evaluating the drug’s effect on ACR in SCD patients (NCT06286046).

### 4.2. Etavopivat

Etavopivat (FT-4202), like mitapivat, is an investigational, oral drug shown to increase hemoglobin’s affinity to oxygen via the activation of the erythrocyte pyruvate kinase (PKR) [47]. The in vivo and ex vivo treatment of sickled RBCs with etavopivat showed a significant increase in Hb oxygen saturation [48]. A randomized, placebo-controlled, double-blind, multicenter phase 2/3 study is currently recruiting pediatric and adult SCD patients (12 to 65 years of age), with the primary endpoint of hemoglobin response and rate of VOC (NCT04624659). Another phase 2 study is evaluating etavopivat in patients with beta-thalassemia and SCD (NCT04987489). The study measures the proportion of patients with a 20% reduction in blood transfusions and is scheduled to be completed in late 2025. Etavopivat is also under investigation for its ability to help SCD patients who are at increased risk of stroke. A phase 2 trial began in July 2023 and is measuring the change in average-mean-maximum-velocity (TAMMV) arterial cerebral blood flow versus baseline (NCT05953584). Future clinical trials on etavopivat aim to study its effect on cerebral hemodynamic response in children with SCD (NCT05725902). The trial was set to begin in March 2024 but has not yet begun recruiting.

## 5. Anti-Adhesion Therapies

### 5.1. Crizanlizumab-tmca (ADAKVEO)

Crizanlizumab is a humanized, anti-P-selectin monoclonal antibody drug administered intravenously to prevent VOCs in SCD patients. In 2019, it was approved by the FDA for patients aged 16 and older and is commercially known as ADAKVEO. The drug stops RBCs, white blood cells, and platelets from adhering to blood vessel walls by binding and inhibiting P-selectin. This also prevents endothelial cells, platelets, sickled RBCs, and leukocytes from binding with each other [49].

The 2016 SUSTAIN (NCT01895361) trial was the primary driving force behind the drug’s approval by the FDA. The trial demonstrated crizanlizumab’s ability to reduce pain crises in SCD patients, with participants reporting 45% fewer pain crises when compared to the placebo group [50]. A multicenter, retrospective, non-interventional study (SUCCESSOR) followed up on the SUSTAIN study participants. Follow-up data report similar results, where VOCs increased after the trial compared to when receiving the intervention [51]. However, beyond this, few studies evaluating its real-world effectiveness have been completed, and many ongoing studies on crizanlizumab continue to investigate its safety and efficacy.

The STAND trial (NCT03814746) is an ongoing phase III trial evaluating the efficacy of two doses of crizanlizumab: 5.0 mg/kg and 7.5 mg/kg. The annualized rate of VOCs leading to a healthcare visit was measured, and 252 participants were split into three experimental groups. Preliminary results released by Novartis are negative, as both the 5.0 mg/kg and 7.5 mg/kg dosages were unable to reduce the annualized rate of VOCs by a statistically significant amount when compared to the placebo [52]. It is important to note that these results have not been published in the peer-reviewed literature and have not been independently reviewed. Therefore, further evaluation is needed before conclusions can be drawn. The study is expected to be completed in 2026.

Because crizanlizumab has only been approved in patients aged 16 years and older, clinical trials investigating its safety and efficacy in pediatric patients are being conducted. The Study of Dose Confirmation and Safety of Crizanlizumab in Pediatric Sickle Cell Disease Patients (NCT03474965) is an ongoing phase II trial investigating the safety and appropriate dosing of crizanlizumab in SCD patients aged 6 months to less than 18 years. The primary outcome measures of the study include the pharmacokinetics of crizanlizumab 15 days after the first dose and at a steady state, along with the frequency of adverse events. The study is scheduled to be completed in 2026, and no preliminary results have been posted.

Several localized studies on crizanlizumab are also being conducted, primarily focusing on the Middle East and India. SPOTLIGHT (NCT05020873) was a multicenter, prospective, single-arm NIS study investigating the effectiveness of crizanlizumab in patients from Middle Eastern countries and India. The study primarily measured the annualized rate of healthcare visits and enrolled 44 participants before being terminated in 2023. Another localized study is the ongoing Indian Multi-centric Phase IV Study to Assess the Safety of Crizanlizumab in Sickle Cell Disease Patients (NCT04662931). This study aims to evaluate the safety of crizanlizumab in Indian SCD patients and primarily measures the frequency, severity, and causality of serious adverse events while being treated with crizanlizumab. A total of 140 participants are enrolled, and the study is scheduled to be completed in 2024.

Finally, crizanlizumab’s ability to improve tissue oxygen supply and treat stroke is under investigation. The CRIZ study (NCT05334576) began in August 2022 and measures the number of new or enlarged silent cerebral infarcts 30 months following treatment. The trial evaluating crizanlizumab’s ability to improve tissue oxygen supply is scheduled to begin in July 2024 and will measure tissue oxygenation by near-infrared spectroscopy.

Overall, crizanlizumab’s effectiveness at treating SCD is mixed, and there are limited data on its effectiveness in patients outside of a clinical trial setting. Despite the success of the original SUSTAIN trial, new clinical studies like STAND do not show the same success. Crizanlizumab may also have significant side effects that limit its viability for many patients. A small single-center analysis of SCD patients was conducted at the University of California San Diego Health from 2020 to 2022 and followed 15 patients who were prescribed crizanlizumab [50]. The researchers found that while crizanlizumab decreased the number of acute care visits the patients made, the discontinuation rate was extremely high, with only one patient remaining on the drug by the end of the observation period. These concerns, along with the poor results of the STAND trial, caused the European Medicines Agency’s Committee for Medicinal Products for Human Use (CHMP) to recommend revoking crizanlizumab’s marketing authorization in Europe. Crizanlizumab’s initial approval by the European Union in 2020 was given on the condition that the STAND trial’s data confirmed the drug’s efficacy. Therefore, in August 2023, the European Commission endorsed CHMP’s recommendation, effectively ending crizanlizumab’s use in the European market [52]. Given this uncertainty and the lack of peer-reviewed data on crizanlizumab, further evaluation of its safety and efficacy is warranted.

### 5.2. Epeleuton

RBCs from individuals with SCD express P-selectin, basal cell adhesion molecule-1/lutheran protein [53], integrin-associated protein (IAP) [54], and vascular cell adhesion molecule (VCAM), which cause RBCs to bind to one another, inflammatory cells such as monocytes and neutrophils, and vascular endothelial layers [55,56,57]. It is believed that the expression of adhesion molecules eventually leads to acute systemic painful VOC. Epeleuton (15-HEPE, 15 hydroxy eicosapentaenoic acid) is a novel, second-generation w-3 fatty acid that targets key drivers of multicellular adhesion and hemolysis [58]. Pre-clinical trials demonstrated epeleuton’s ability to down-regulate several adhesion molecules including VCAM-1, ICAM-1, and E-selectin, making it a potential therapeutic for SCD [59]. In January 2024, a trial measuring the pharmacokinetics, pharmacodynamics, and safety of epeleuton in SCD patients began, enrolling 30 participants (NCT05861453). The primary endpoints of the study include changes in the levels of P-selectin, E-selectin, and VCAM-1 and the annual rate of VOCs. Estimated to be completed in late 2024, this study could establish epeleuton’s potential as an SCD therapy and justify additional clinical trials.

### 5.3. Famotidine

Famotidine, a histamine type 2 receptor antagonist that reduces P-selectin, is being piloted in an interventional study currently recruiting pediatric patients (1 to 17 years of age) with SCD. The planned study will administer 400 mg/50 mL of famotidine, 0.5 mg/kg/12 h (with a maximum dose of 80 mg/day) during a 29-trial period. The primary endpoint is reduced P-selectin levels (NCT05084521).

### 5.4. Inclacumab

Inclacumab is a recombinant, fully human monoclonal antibody that binds to P-selectin, preventing its ligand binding activity, thereby interfering with the adhesion of sickled RBCs, platelets, and leukocytes to the endothelium, with the long-term goal of reducing VOCs [60]. Two phase 3 multicenter, open-label, randomized double-blind studies are currently investigating the efficacy, safety, and long-term treatment of VOC with inclacumab in adults and youths (NCT04935879 and NCT04927247). An extension study for patients who have already completed a prior study of inclacumab is ongoing, measuring treatment adverse events in 520 participants (NCT05348915).

### 5.5. Intravenous Gamma Globulin (IVIG)

In preclinical and clinical studies, IVIG was found to inhibit the adhesion of neutrophils to the endothelium and RBCs [61,62]. IVIG is currently in a phase II randomized clinical trial in which a 400 mg/kg dose is administered to adult and children SCD patients with VOCs (12 to 65 years old; NCT01757418). In a midpoint analysis of this trial, Manwani et al. (2020) reported that neutrophil activation was significantly improved 24 h after IVIG administration compared to the placebo group. In the younger age group (<14 years old), the median length of VOC was 59.65 h, compared to 78.30 h for the control group. No significant changes were reported in the older age group [63].

## 6. Gene Therapy

Gene therapy is being extensively researched as a potential cure for SCD. Advances in genomic sequencing have made it possible to understand Hb regulation, and discoveries in the genome modification of hematopoietic stem cells serve as possible alternative cures for SCD. CD34+ is a cell surface marker typically utilized to identify hematopoietic and progenitor stem cells in bone marrow transplant studies to develop targeted and patient-centric genetic therapies [64]. Gene-addition mechanisms using gene transfer vectors (lentiviral vectors or LV) have been optimized over the past few decades to increase the expression of normal or antisickling globins as strategies to treat SCD [65,66]. For instance, genome association studies have discovered that the BCL11A gene is directly involved in HbF production, and knocking down the gene increases HbF levels [67]. Gene therapy has also been found to lower the risk of SCD complications such as VOCs, acute chest syndrome, and GVHD associated with HSCT. Below, we review clinical studies on some of the most promising gene therapeutics, with two recently being FDA-approved to treat SCD patients.

### 6.1. BCH-BB694

BCH-BB694 is a lentiviral vector that encodes a miRNA-adapted short hairpin RNA (shRNAmiR) targeting BCL11A. In an ongoing open-label, non-randomized, single-center, single-arm cohort study (NCT03282656), autologous bone marrow-derived CD34+ HSC cells were transduced with BCH-BB694 to knock down the BCL11A protein [68]. Seven SCD patients (ages 12–25 at enrollment) then received a single transfusion of edited cells, followed by neutrophil and platelet engraftment. HbF and HbA levels were normalized and remained stable. As of their latest publication, none of the patients experienced any VOCs, acute chest syndrome, or stroke following the gene therapy [68]. Another phase 2 clinical study called GRASP (NCT05353647) is also investigating BCH-BB694 and its ability to eliminate VOEs in patients with SCD. The study classifies success as the complete absence of VOEs 24 months after treatment. The study began in 2022 and is expected to be completed in May 2025.

### 6.2. Casgevy/Exa-Cel (CTX001)

Casgevy (CTX001) is an autologous gene therapy based on CD34+ hematopoietic stem and progenitor cells (HSPCs). The HSPCs were edited with clustered regularly interspaced short palindromic repeats (CRISPR)-Cas9 targeting the BCL11A protein. Frangoul et al. examined the administration of CTX001 in two patients, one with SCD and the other with transfusion-dependent β-thalassemia (TDT). The study successfully demonstrated the therapy’s effectiveness in significantly increasing HbF levels in SCD patients. Similar results were reported in a follow-up study of eight additional adult patients (six with TDT and two with SCD) [69]. In 2018, a single-arm, open-label, multi-site, single-dose phase 1/2/3 study on CTX001 was conducted in patients with severe SCD (NCT03745287). In 2024, CTX001 became the first CRISPR drug to be approved by the FDA. Currently, clinical studies are focusing on studying the efficacy and safety of Casgevy in patients (NCT05477563; NCT05329649). Future clinical trials on Casgevy will continue to evaluate its safety in patients with SCD. One trial is set to begin in April 2024 and will evaluate Casgevy in patients with HbSC genotype SCD (NCT05951205). The primary outcome of the study is the percentage of patients that achieved fetal hemoglobin percentages of greater than 20%.

### 6.3. Lyfgenia/Lov-Cel (BB305)

BB305 is a globin lentiviral vector encoded with a healthy β-globin gene that codes for antisickling hemoglobin, HbAT87Q. In HbAT87Q, the amino acid threonine is substituted with glutamine at the 87th position, enabling it to inhibit the sterical polymerization of sickle hemoglobin [70]. Magrin et al. examined the efficacy of BB305 (also called b1111 or LentiGlobin) as an antisickling gene therapy. In an open-label, single-arm, non-randomized 2-year interventional study, followed by an observational study in three patients ranging in age from 13 to 21 years old, hematopoietic stem cells derived from the patient’s bone marrow were enriched with CD34+ cells transduced with BB305 (LentiGlobin) [66,71]. Remission of SCD was observed in two patients without recurring VOCs or acute chest syndrome [71]. Pain medications for SCD-related complications were discontinued, and transfusion therapy was no longer needed. However, one patient experienced acute chest syndrome and VOCs.

Another multicenter, non-randomized, open-label, single-dose phase 1–2 clinical trial (NCT02140554) was conducted at 11 sites across the United States. In this trial, researchers transplanted autologous hematopoietic stem and progenitor cells transduced with BB305 lentiviral vector into 35 SCD patients (12 to 50 years of age). The patients were followed for approximately 17 months [70]. Each patient received a transfusion of healthy erythrocytes to support the safe and optimal proliferation of the engraftment and to prevent any side effects. Of the 35 patients, 25 were evaluated and found to have stable productions of HbAT87Q and no severe VOCs. BB305 is currently being investigated in 35 adult and pediatric SCD patients in a phase 3 non-randomized trial, with the primary outcome of complete resolution of VOCs between 6 and 18 months of therapy infusions (NCT04293185). BB305 (or LentiGlobin/bb1111) was approved by the FDA in 2024 and is now also known as Lyfgenia.

### 6.4. EDIT-301

EDIT-301 is an experimental autologous cell therapy that increases HbF production by utilizing CRISPR-Cas12a to modify CD34+ cells. Previous research on patients with hereditary persistence of fetal hemoglobin (HPFH) found that elevated HbF levels were often associated with the disruption of a distal CCAAT box region on the HBG1/2 promoters [72]. EDIT-301 targeted this region and was able to produce productive indel mutations in 80–90% of CD34+ cells from a healthy donor. These edited cells were able to produce 40% HbF expression in their erythroid progeny and maintained this level when infused into NBSGW mice [72]. In 2021, EDIT-301 began testing in humans with the RUBY trial (NCT04853576), a phase 1/2 study evaluating the safety and efficacy of EDIT-301 in patients with severe SCD. Patients were given a single dose of hematopoietic stem cells edited with EDIT-301, and the proportion of subjects who achieved the complete resolution of VOEs was measured. Preliminary results were promising, as the first two study participants were able to achieve an editing rate of 80% or higher in CD34+ and peripheral blood nucleated cells [73]. Eight months after treatment, the first subject had an F-cell percentage of 96.5%, and all markers of hemolysis improved or normalized. Additionally, no VOEs or treatment-related adverse events were reported in the four participants who had received EDIT-301 [73]. These preliminary results are quite limited, however, as 2 of the 4 participants had received the therapy less than a month prior, and the RUBY study enrolled a total of 45 participants. The study is expected to be completed in August 2025.

### 6.5. BEAM-101

BEAM-101, like EDIT-301, is an experimental autologous HSC therapy that targets the HBG1/2 promoter. BEAM-101 utilizes CRISPR-based adenine base editors to induce point mutations in the HBG1/2 promoter, resulting in increased HbF production. In preclinical mouse models, greater than 90% of edited HSCs were stably edited and resulted in over 65% HbF expression [74]. In 2022, the BEACON study began, evaluating the safety and efficacy of BEAM-101 in patients with SCD. The primary endpoints include the change in the annual number of VOCs, the proportion of patients with successful neutrophil engraftments, transplant-related mortality within 100 days, and the frequency of adverse events. In January 2024, the first patient for the BEACON trial was dosed with BEAM-101, with clinical data expected in the second half of 2024.

## 7. Other Novel Therapeutics to Treat SCD Complications

### 7.1. L-Glutamine (Endari)

Glutamine is a conditionally essential amino acid involved in nitrogen transport and serves as a precursor for synthesizing other biological molecules, such as glutathione, nicotinamide adenine dinucleotide (NAD), and arginine. Reactive oxygen species (ROS) are produced naturally via enzymatic or non-enzymatic processes. They are neutralized in mammalian cells by antioxidant pathways that involve glutathione (GSH), NAD (H), NADP (H), and nitric oxide (NO). Sickled RBCs have higher levels of ROS, causing damage to the RBC membrane and contributing to the pathophysiology of SCD. Therapeutic interventions that induce the production of HbF such as HU or ones that increase substrates for GSH, NADH, NADPH, and NO counteract ROS activity [8]. The biochemical mechanism of glutamine in SCD is currently not well understood.

The oral administration of glutamine potentially protects sickled RBCs from hemolysis and oxidative damage by increasing NAD redox ratios within the sickle cell [75,76,77]. In a phase III trial, Niihara et al. conducted a 2:1 randomized, double-blind, multicenter study on 230 patients (ages 5–58) with sickle cell anemia and sickle ß0-thalassemia (NCT01179217). Patients were administered 0.3 g/kg of L-glutamine twice daily for 48 weeks. Two-thirds of the patients in the placebo and L-glutamine groups were also administered HU. Patients on L-glutamine had lower incidences of hospitalizations and fewer reports of pain crises with or without hydroxyurea compared to the placebo group [77]. Currently, the only clinical trial on L-glutamine is a phase IV study to determine the safety and efficacy of L-glutamine in preventing VOC in pediatric and adult SCD patients (NCT05371184).

### 7.2. Defibrotide

Defibrotide is a polydisperse oligonucleotide originally approved to treat hepatic vaso-occlusive disease following HSCT. Studies are currently underway to determine its effectiveness in managing VOC in SCD patients [78]. Defibrotide was found to be safe and well tolerated in SCD patients (2 to 35 years of age) with a high risk of stroke, acute chest syndrome, pain crises, abnormal transcranial Doppler ultrasound (TCD) values, and silent infarct lesions. All patients received myeloimmunoablative conditioning (MAC) and haploidentical stem cell transplantation utilizing CD34+ selection and T-cell (CD3) addback. Preliminary data from a 2020 phase II clinical trial (IND 127812) showed defibrotide was well tolerated without severe adverse events or bleeding events reported [79]. Of the thirteen patients enrolled in the trial, all but one patient had the resolution of their fever. Patients required an average of 1.15 days of oxygen support, with no patients requiring mechanical ventilation [80]. Defibrotide is currently being investigated for safety in a phase II open-label clinical trial, where 6.25 mg/kg is administered in patients with SCD and related acute chest syndrome (2 years to 40 years of age; NCT03805581). Primary outcome endpoints are allergic reactions to defibrotide and grade III/IV bleeding within 30 days of administration. The secondary outcome is improved clinical signs of acute chest syndrome within 30 days of administration.

### 7.3. ALXN1820

The complement system is a key component of the immune system and induces a series of inflammatory responses to help fight infection. However, recent studies have found that increased complement activation via the alternate pathway has been linked to SCD complications including VOCs, hemolysis, inflammation, end-organ damage, and delayed hemolytic transfusion reaction (DHFR) [81,82]. In particular, defective regulation of the C5b-9 membrane attack complex is correlated with a major increase in sickle cell lysis [82]. This makes the complement system a potential therapeutic target for SCD. ALXN1820 is a humanized, bispecific, VHH (variable heavy domain) antibody that simultaneously binds human albumin and properdin [83]. The antibody selectively inhibits CAP (complement alternate pathway) activation and has an extended circulatory half-life due to its binding to albumin. Mouse models have demonstrated that properdin inhibition results in significantly reduced VOCs and hemolysis, making ALXN1820 a promising new therapy [83]. The PHOENIX trial (NCT05565092) is an ongoing phase 2a trial investigating the safety and efficacy of ALXN1820 in adults with SCD. The primary endpoint of the study is the number of emergent treatment adverse effects and serious adverse effects, but pharmacokinetics, changes in baseline complement activation, hemoglobin, and hemolysis will also be measured. The study is scheduled to be completed in June 2024.

### 7.4. Crovalimab

Crovalimab is an anti-C5 sequential monoclonal antibody that was originally designed to treat paroxysmal nocturnal hemoglobinuria but is also under investigation as a potential adjunct therapy for SCD. The drug binds to the beta chain of C5, increasing C5 uptake by endothelial cells and preventing C5 convertase from activating the protein. This inhibits C5b6 deposition on cell membranes and limits membrane attack complex tissue damage [84]. Two clinical trials called CROSSWALK-a (NCT04912869) and CROSSWALK-c (NCT05075824) began in 2022 and are investigating the safety and efficacy of crovalimab in SCD patients. CROSSWALK-a focuses on the safety of crovalimab, and its primary endpoints are the percentage of patients who experienced adverse events and infusion-related reactions. CROSSWALK-c focuses on the efficacy of crovalimab as an adjunct SCD therapy, and its primary endpoint is the annual rate of medical facility VOCs. Both trials are expected to end in 2025.

### 7.5. RNA Aptamers

Aptamers are short, single-stranded oligonucleotides that are capable of binding molecular targets with high affinity and specificity due to their ability to fold into unique structures. They are also nonimmunogenic, nontoxic, and far cheaper than gene therapy, giving them significant therapeutic potential [85]. Purvis et al. showed that two RNA aptamer molecules (OX3B and DE3A) bind to sickle hemoglobin, preventing polymerization without affecting the ability of the red cells to carry oxygen [85]. These studies are ongoing. Additionally, an aptamer (ARC5690) that targets P-selectin was investigated in SCD model mice where it inhibited the adhesion of sickle red cells and leukocytes to endothelial cells [86]. These aptamers are in preclinical development, and no studies with human participants are currently ongoing.

### 7.6. Nitric Oxide (NO) Modulation

NO is a highly reactive signaling molecule that is important in vascular hemostasis [87,88]. NO is synthesized by distinct subtypes of NO synthase (NOS) from the catalysis of L-arginine, utilizing electrons from NADPH and producing L-citrulline in the process [87]. It is hypothesized that depleted levels of NO lead to SCD-related organ injury, VOC, and acute pain episodes [88].

In SCD patients, damage to endothelial layers caused by VOEs triggers hemolysis and the inflammatory response, leading to the release of the enzyme arginase. This enzyme degrades L-arginine, a substrate involved in NO synthesis, lowering NO production [87,88]. Additionally, asymmetric dimethyl arginine (ADMA) levels are elevated, which are involved in the metabolism of L-arginine and inhibit NOS, further decreasing NO levels [57,87]. One way to overcome NO deficiency is with L-arginine administration, which serves as a nitrogen donor during NO synthesis. Two randomized studies are currently recruiting SCD pediatric and adult patients to determine the safety and efficacy of L-arginine to treat NO deficiency with Increased Tricuspid Regurgitant Jet Velocity or TRJV (NCT02447874 and NCT05470998). TRJV is measured to determine pulmonary artery pressure in SCD patients. Elevated TRJV levels (>2.5 m/s) are associated with pulmonary hypertension in adults and have been found to serve as precursors to pulmonary hypertension in pediatric patients [89]. Primary outcome measures evaluate changes in NO metabolites and pharmacokinetics of IV arginine, measured by plasma arginine concentration over time. Additional outcome measures for the NCT02447874 study include determining whether giving extra arginine will ameliorate vaso-occlusive painful event (VOE) scores, decrease the need for pain medications, or decrease the length of a hospital stay or emergency department visit.

### 7.7. Alternatives to Drug Therapies

Several studies are underway to determine the efficacy of alternative treatments to alleviate pain in SCD patients. They include acupuncture analgesia, yoga and meditation, exercise [90,91], virtual reality, and hypnotic script virtual reality pain management. Low levels of omega-3 fatty acids, docosahexaenoic acid (DHA), and eicosapentaenoic acid (EPA) have been associated with hs-CRP and inflammation in SCD [92]. A comparative analysis of the effectiveness of omega-3 fatty acid or vitamin-D supplementation along with standard of care (hydroxyurea, folic acid, and ibuprofen) was conducted in pediatric SCD patients (age 7 to 18 years of age) in Beni Suef and Giza, Egypt. The group that received omega-3 fatty acid with the standard of care had markedly reduced pain episodes compared to the other groups [93]. Similarly, studies reported lower levels of VOCs and hospitalizations [94]. This treatment also appears to be more cost-effective than standard care levels [93]. Randomized clinical trials to study the efficacy and safety of omega-3 fatty acids and/or vitamin-D supplementation are underway (NCT02947100 and NCT04662476). A study examining the pharmacokinetics and pharmacodynamics of Ilera Medical Marijuana products in various diseases, including SCD, was terminated due to feasibility issues (NCT03886753).

## 8. Conclusions

In general, treatments for patients with SCD focus on managing symptoms, preventing complications, and improving the patient’s overall quality of life. While the only cure for SCD is bone marrow transplantation, recently, the FDA approved two gene therapies (Casgevy and Lyfgenia) for treating patients with SCD (Figure 3). Indeed, the clinical trials are promising, as patients who received these treatments reported a significant reduction in or total elimination of pain. While this is inspiring and promising news, there is still a need for more effective therapies. One of the significant drawbacks of these gene therapies is the cost, which ranges in the millions, making it nearly impossible to cure all of the individuals with SCD. This review article summarizes some of the ongoing or recently completed clinical trials addressing various aspects of SCD, from the new gene therapy approaches to various drugs to manage pain and vas-occlusive complications. Based on the number of trials, it is clear that discovering effective treatments or “cures” for SCD remains essential. However, despite the number of individuals with this debilitating disease, the research funding for SCD is disproportionately low when taking into account disease burden and disease prevalence. Nevertheless, there is hope, and with continual clinical and basic science research, an affordable “cure” is within reach.

## Figures and Tables

**Figure 1 cells-13-00848-f001:**
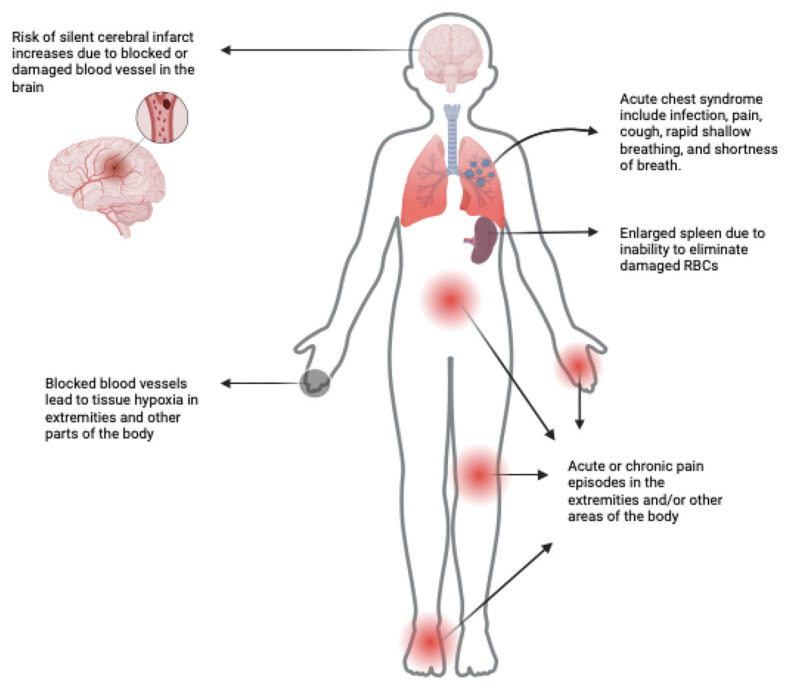
Complications of SCD disease.

**Figure 2 cells-13-00848-f002:**
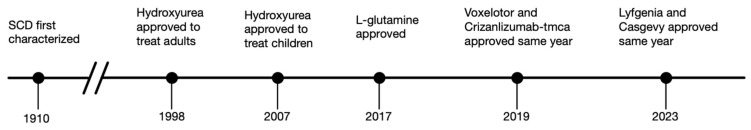
Timeline of FDA-approved SCD treatments.

**Figure 3 cells-13-00848-f003:**
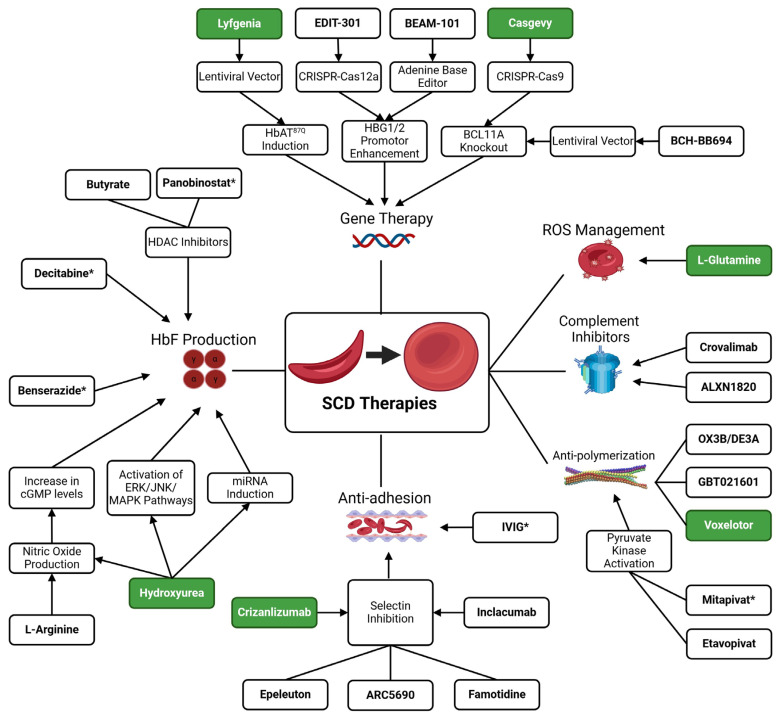
Approved and novel therapies and the associated pathways they affect in the treatment of SCD and its complications. The shaded boxes represent FDA-approved therapies, namely hydroxyurea, L-glutamine, voxelotor, crizanlizumab, Lyfgenia, and Casgevy. Boxes with asterisks are drugs that are FDA-approved, but not for treating SCD. The remaining boxes include some novel therapies.

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
