# Peer review of "Current and Future Therapeutics for Treating Patients with Sickle Cell Disease"

_cells, 2024, doi:10.3390/cells13100848_

Round 1

Reviewer 1 Report

Comments and Suggestions for Authors

The article reviews  “Current and future therapeutics for treating patients with sickle cell disease.” Overall, the authors attempt to present as comprehensive a coverage as possible and this is a very challenging task, a result of this is that the review is very skewed with certain areas dealt in great detail, and others, eg, another pyruvate kinase activator (mitapivat) which is far more advanced than etavopivat has been completely left out.  I also think the review needs structuring, perhaps along the lines of therapeutic targets – Curative , targeting HbS polymerisation, and then the downstream targets. Curative therapies have been extensively reviewed and this review does not do its justice. It is probably worth a mention and then cite recent reviews in this area. 

This review is comprehensive in the inclusion of emerging clinical trials for agents to treat sickle cell disease. In particular, the sections on voxelotor, GBT021601, and L-glutamine are well researched. The text is easy to understand. However, the manuscript does not review the foundational data of current or future therapeutics to the same degree as voxelotor.  Including this data would help the reader contextualize the significance of these therapeutic agents.

For example, in the hydroxyurea (HU) section, the current manuscript does not highlight the foundational nature of hydroxyurea and how it ameliorates much of the clinical syndrome of sickle cell disease including the reduction in crisis incidence and intensity, the increase in total hemoglobin, and the reduction of the risk of developing acute chest syndrome most strongly supported by the Multicenter Study of Hydroxyurea trial (MSH).1 Hydroxyurea was then shown to reduce mortality compared to patients not taking HU even in long term follow up studies.2,3 Hydroxyurea is known to inhibit ribonucleotide reductase, which is thought to favor production of cells that divide less rapidly (HbF versus HbS). 4 These mechanisms are not included in the text but are included in a subsequent table.  In addition, one of the major concerns for long term use of hydroxyurea is long-term safety – in particular unanswered questions about fertility, ulcers and malignancy risk. There is an open label extension study (ESCORT-HU) which is looking at this question (NCT04707235) which is included in the table but not in the text. This would help the reader understand why Hydroxyurea is so important relative to Metformin, both which are included under the same heading. It would also contextualize why Decitabine has been studied again in small Phase I/II trials as an alternative to hydroxyurea although it was first discovered to induce hemoglobin F in the 1980s.5,6 The trial data for Crizanlizumab is complete, but again did not contextualize that in the one-year follow-up of the Phase III trial (STAND), the primary end-point of reduction of VOC did not reach statistical significance thus prompting the European Union to withdraw its approval.Aong thelines of HbF induction, an important agent under evaluation is Bensarazide (NCT04432623), and this is not mentioned at all.

There may be some opportunities to consider the re-organization of the article. Adding subsections after mechanism of action to subdivide into which agents are FDA approved for sickle cell, which are FDA approved in general but not for sickle cell, and then those that are completely experimental therapies with those with positive or negative outcomes can help organize the sub-sections. This same organization can be applied to the table 1-4, or organizing by the trials by their Stages.

Pyruvate kinase activators should be included as a separate category apart from anti-sickling agents as in their mechanism of action they not only prevent sickling, but by increasing cellular hydration and intracellular ATP they may reduce ineffective erythropoiesis and may reduce crisis.8 Mitavipat was also not included. Other novel therapeutic agents in Phase II or III clinical trials include complement inhibition with Crovalimab (NCT05075824) and ALXN1820 SC (NCT05565092) particularly given the relationship between complement activation with delayed hemolytic transfusion reactions (DHTR).9 The ferroportin inhibitor Vamifeport (NCT04817670) in Phase II trials should also be considered given the association between circulating free iron and its deleterious effects causing sterile inflammation, endothelial dysfunction and promotes sickle complications.10 

In summary:

1.     This review needs a major re-structuring.

2.     The table is very confusing and unstructured. This can either be organized as FDA-approved agents, or under pathways of targeting.

References

1.     Charache S, Terrin ML, Moore RD, Dover GJ, Barton FB, Eckert SV, McMahon RP, Bonds DR. Effect of hydroxyurea on the frequency of painful crises in sickle cell anemia. Investigators of the Multicenter Study of Hydroxyurea in Sickle Cell Anemia. N Engl J Med. 1995 May 18;332(20):1317-22. doi: 10.1056/NEJM199505183322001. PMID: 7715639.

2.     Steinberg MH, McCarthy WF, Castro O, Ballas SK, Armstrong FD, Smith W, Ataga K, Swerdlow P, Kutlar A, DeCastro L, Waclawiw MA; Investigators of the Multicenter Study of Hydroxyurea in Sickle Cell Anemia and MSH Patients' Follow-Up. The risks and benefits of long-term use of hydroxyurea in sickle cell anemia: A 17.5 year follow-up. Am J Hematol. 2010 Jun;85(6):403-8. doi: 10.1002/ajh.21699. PMID: 20513116; PMCID: PMC2879711.

3.     Platt OS, Brambilla DJ, Rosse WF, Milner PF, Castro O, Steinberg MH, Klug PP. Mortality in sickle cell disease. Life expectancy and risk factors for early death. N Engl J Med. 1994 Jun 9;330(23):1639-44. doi: 10.1056/NEJM199406093302303. PMID: 7993409.

4.     Elford HL. Effect of hydroxyurea on ribonucleotide reductase. Biochem Biophys Res Commun. 1968 Oct 10;33(1):129-35. doi: 10.1016/0006-291x(68)90266-0. PMID: 4301391.

5.     Koshy M, Dom L, Bressler L, Molokie R, Lavelle D, Talischy N, Hoffman R, van Overveld W, DeSimone J. 2′-deoxy-5-azacytidine and fetal hemoglobin induction in sickle cell anemia. Blood. 2000;96:2379–2384

6.     Saunthararajah Y, Molokie R, Saraf S, Sidhwani S, Gowhari M, Vara S, Lavelle D, DeSimone J. Clinical effectiveness of decitabine in severe sickle cell disease. Br J Haematol. 2008 Apr;141(1):126-9. doi: 10.1111/j.1365-2141.2008.07027.x. PMID: 18324975.

7.     European Commission (EC) adopts decision endorsing CHMP recommendation to revoke the conditional marketing authorization for Adakveo® (crizanlizumab) | Novartis

8.     Dina S. ParekhWilliam A. EatonSwee Lay Thein; Recent developments in the use of pyruvate kinase activators as a new approach for treating sickle cell disease. Blood 2024; 143 (10): 866–871. doi: https://doi.org/10.1182/blood.2023021167

9.     Varelas C, Tampaki A, Sakellari I, Anagnostopoulos Α, Gavriilaki E, Vlachaki E. Complement in Sickle Cell Disease: Are We Ready for Prime Time? J Blood Med. 2021 Mar 23;12:177-187. doi: 10.2147/JBM.S287301. PMID: 33790681; PMCID: PMC8001680.

10.  Gbotosho OT, Kapetanaki MG, Kato GJ. The Worst Things in Life are Free: The Role of Free Heme in Sickle Cell Disease. Front Immunol. 2021 Jan 27;11:561917. doi: 10.3389/fimmu.2020.561917. PMID: 33584641; PMCID: PMC7873693.

Author Response

Response to Reviewer 1

Summary

The authors would like to thank you very much for taking the time to review this manuscript. Please find the detailed responses below and the corresponding revisions/corrections highlighted/in track changes in the re-submitted files. Based on your comments, the authors feel that the article is much improved.

Point by point responses

The article reviews  “Current and future therapeutics for treating patients with sickle cell disease.” Overall, the authors attempt to present as comprehensive a coverage as possible and this is a very challenging task, a result of this is that the review is very skewed with certain areas dealt in great detail, and others, eg, another pyruvate kinase activator (mitapivat) which is far more advanced than etavopivat has been completely left out. 

This was an oversight on our part. The authors have now included a section on mitapivat (pages 6 and 7), as suggested by this reviewer and others. We agree that it is more advanced than etavopivat; however, we kept the section on etavopivat because of some of the recent clinical trial data.

 I also think the review needs structuring, perhaps along the lines of therapeutic targets – Curative , targeting HbS polymerisation, and then the downstream targets. Curative therapies have been extensively reviewed and this review does not do its justice. It is probably worth a mention and then cite recent reviews in this area. 

The authors have restructured the article, focusing on the ongoing studies and some new studies that are approved but have not started recruiting participants. The authors acknowledged the previous reviews, particularly ones that focus on SCD therapeutics and gene therapy for example, (Kuriri, FA, 2024, and White, SL et al., 2023); however, our review is different since we focus on clinical studies, in addition to some recent studies, and novel therapeutics. Figure 3 is particularly relevant, as we highlighted the FDA-approved drugs; however, we have now included drugs that are FDA-approved but not for treating sickle cell patients.

This review is comprehensive in the inclusion of emerging clinical trials for agents to treat sickle cell disease. In particular, the sections on voxelotor, GBT021601, and L-glutamine are well researched. The text is easy to understand. However, the manuscript does not review the foundational data of current or future therapeutics to the same degree as voxelotor.  Including this data would help the reader contextualize the significance of these therapeutic agents.

The authors agree with the comment and have included foundational data on current and future therapeutics. For example, the authors have a section on nucleic acids and how they could potentially be used as effective therapeutics in the future, this has been incorporated into the section on anti-sickling. The authors have added more background information on the various therapeutics.

For example, in the hydroxyurea (HU) section, the current manuscript does not highlight the foundational nature of hydroxyurea and how it ameliorates much of the clinical syndrome of sickle cell disease including the reduction in crisis incidence and intensity, the increase in total hemoglobin, and the reduction of the risk of developing acute chest syndrome most strongly supported by the Multicenter Study of Hydroxyurea trial (MSH).1 Hydroxyurea was then shown to reduce mortality compared to patients not taking HU even in long term follow up studies.2,3 Hydroxyurea is known to inhibit ribonucleotide reductase, which is thought to favor production of cells that divide less rapidly (HbF versus HbS). 4 These mechanisms are not included in the text but are included in a subsequent table.  In addition, one of the major concerns for long term use of hydroxyurea is long-term safety – in particular unanswered questions about fertility, ulcers and malignancy risk. There is an open label extension study (ESCORT-HU) which is looking at this question (NCT04707235) which is included in the table but not in the text. This would help the reader understand why Hydroxyurea is so important relative to Metformin, both which are included under the same heading.

The authors agree with the reviewer. We have included this study in the text; however, we initially chose not to discuss the foundational contents of HU, as that has been discussed previously. However, based on the reviewer’s suggestion, we have returned this content back into the text.

 It would also contextualize why Decitabine has been studied again in small Phase I/II trials as an alternative to hydroxyurea although it was first discovered to induce hemoglobin F in the 1980s.5,6 The trial data for Crizanlizumab is complete, but again did not contextualize that in the one-year follow-up of the Phase III trial (STAND), the primary end-point of reduction of VOC did not reach statistical significance thus prompting the European Union to withdraw its approval.Aong thelines of HbF induction, an important agent under evaluation is Bensarazide (NCT04432623), and this is not mentioned at all.

The authors agree and have included each of the drugs discussed in a proper context. 

There may be some opportunities to consider the re-organization of the article. Adding subsections after mechanism of action to subdivide into which agents are FDA approved for sickle cell, which are FDA approved in general but not for sickle cell, and then those that are completely experimental therapies with those with positive or negative outcomes can help organize the sub-sections. This same organization can be applied to the table 1-4, or organizing by the trials by their Stages.

The authors have reorganized the article and have removed the tables (as suggested by other reviewers). The authors feel strongly that having the manuscript organized by mechanism allows the reader to contextualize the advantages of some of these drugs compared to others. Thus, putting them in the same category is required.  Also, we have indicated the FDA approved drugs for SCD and the FDA approved drugs in general in Figure 3.

Pyruvate kinase activators should be included as a separate category apart from anti-sickling agents as in their mechanism of action they not only prevent sickling, but by increasing cellular hydration and intracellular ATP they may reduce ineffective erythropoiesis and may reduce crisis.8 

The authors agree and have removed the pyruvate kinase activators from the anti-sickling section, and have put it into its own section.

Mitavipat was also not included. Other novel therapeutic agents in Phase II or III clinical trials include complement inhibition with Crovalimab (NCT05075824) and ALXN1820 SC (NCT05565092) particularly given the relationship between complement activation with delayed hemolytic transfusion reactions (DHTR).9 The ferroportin inhibitor Vamifeport (NCT04817670) in Phase II trials should also be considered given the association between circulating free iron and its deleterious effects causing sterile inflammation, endothelial dysfunction and promotes sickle complications.10 

The authors would like to thank the reviewer for this comment. These novel therapeutics are now included in the text and the figure. We feel that having the novel therapeutics in this review makes it a much more substantial article.

In summary:

  1. This review needs a major re-structuring.

The authors have restructured the review based on the comments from all three reviewers and now feel that it is much better.

  1. The table is very confusing and unstructured. This can either be organized as FDA-approved agents, or under pathways of targeting.

The authors have removed most of the tables, as most of the studies are in the text. The tables, while necessary, were redundant.

References

  1. Charache S, Terrin ML, Moore RD, Dover GJ, Barton FB, Eckert SV, McMahon RP, Bonds DR. Effect of hydroxyurea on the frequency of painful crises in sickle cell anemia. Investigators of the Multicenter Study of Hydroxyurea in Sickle Cell Anemia. N Engl J Med. 1995 May 18;332(20):1317-22. doi: 10.1056/NEJM199505183322001. PMID: 7715639.
  2. Steinberg MH, McCarthy WF, Castro O, Ballas SK, Armstrong FD, Smith W, Ataga K, Swerdlow P, Kutlar A, DeCastro L, Waclawiw MA; Investigators of the Multicenter Study of Hydroxyurea in Sickle Cell Anemia and MSH Patients' Follow-Up. The risks and benefits of long-term use of hydroxyurea in sickle cell anemia: A 17.5 year follow-up. Am J Hematol. 2010 Jun;85(6):403-8. doi: 10.1002/ajh.21699. PMID: 20513116; PMCID: PMC2879711.
  3. Platt OS, Brambilla DJ, Rosse WF, Milner PF, Castro O, Steinberg MH, Klug PP. Mortality in sickle cell disease. Life expectancy and risk factors for early death. N Engl J Med. 1994 Jun 9;330(23):1639-44. doi: 10.1056/NEJM199406093302303. PMID: 7993409.
  4. Elford HL. Effect of hydroxyurea on ribonucleotide reductase. Biochem Biophys Res Commun. 1968 Oct 10;33(1):129-35. doi: 10.1016/0006-291x(68)90266-0. PMID: 4301391.
  5. Koshy M, Dom L, Bressler L, Molokie R, Lavelle D, Talischy N, Hoffman R, van Overveld W, DeSimone J. 2′-deoxy-5-azacytidine and fetal hemoglobin induction in sickle cell anemia. Blood. 2000;96:2379–2384
  6. Saunthararajah Y, Molokie R, Saraf S, Sidhwani S, Gowhari M, Vara S, Lavelle D, DeSimone J. Clinical effectiveness of decitabine in severe sickle cell disease. Br J Haematol. 2008 Apr;141(1):126-9. doi: 10.1111/j.1365-2141.2008.07027.x. PMID: 18324975.
  7. European Commission (EC) adopts decision endorsing CHMP recommendation to revoke the conditional marketing authorization for Adakveo® (crizanlizumab) | Novartis
  8. Dina S. Parekh, William A. Eaton, Swee Lay Thein; Recent developments in the use of pyruvate kinase activators as a new approach for treating sickle cell disease. Blood2024; 143 (10): 866–871. doi: https://doi.org/10.1182/blood.2023021167
  9. Varelas C, Tampaki A, Sakellari I, Anagnostopoulos Α, Gavriilaki E, Vlachaki E. Complement in Sickle Cell Disease: Are We Ready for Prime Time? J Blood Med. 2021 Mar 23;12:177-187. doi: 10.2147/JBM.S287301. PMID: 33790681; PMCID: PMC8001680.
  10. Gbotosho OT, Kapetanaki MG, Kato GJ. The Worst Things in Life are Free: The Role of Free Heme in Sickle Cell Disease. Front Immunol. 2021 Jan 27;11:561917. doi: 10.3389/fimmu.2020.561917. PMID: 33584641; PMCID: PMC7873693.

The authors would like thank this reviewer for the critiques and the references, which are incorporated into our review article.

Reviewer 2 Report

Comments and Suggestions for Authors

The manuscript is a comprehensive revision of ongoing and recently concluded clinical trials (2020 to 2024)  in patients with sickle cell disease. The revision mainly focused on therapeutics for managing the clinical course of SCD patients and potential cures for this disease.

Overall, the review is clearly written.  However, I have some concerns. 

The authors grouped clinical trials according to the mechanism of action of the investigated treatment (HbF induction, anti-polymerization, anti-adhesion, gene therapies, and others). Considering that not all studies listed in the Tables are mentioned or discussed in the text, Tables (and supplementary tables) should give more information,  including at least the primary study outcome, and, for completed studies, the main findings.

There is no specific paragraph on HSCT, but there is a supplementary table listing several studies in this setting. The authors should explain why they decided not to comment on HSCT studies.   

 Line 497:  the authors cited Table 6, but it does not exist

Author Response

Response to Reviewer 2

Summary

The authors would like to thank you very much for taking the time to review this manuscript. Please find the detailed responses below and the corresponding revisions/corrections highlighted/in track changes in the re-submitted files. Based on your comments, the authors feel that the article is much improved.

Point by point responses

The manuscript is a comprehensive revision of ongoing and recently concluded clinical trials (2020 to 2024)  in patients with sickle cell disease. The revision mainly focused on therapeutics for managing the clinical course of SCD patients and potential cures for this disease.

Overall, the review is clearly written.  However, I have some concerns. 

The authors grouped clinical trials according to the mechanism of action of the investigated treatment (HbF induction, anti-polymerization, anti-adhesion, gene therapies, and others). Considering that not all studies listed in the Tables are mentioned or discussed in the text, Tables (and supplementary tables) should give more information,  including at least the primary study outcome, and, for completed studies, the main findings.

The authors agree with this comment.  We have removed the tables, instead we have included the studies in the text that are more pertinent to the objective to the review article.  We agree, that some of the studies in the tables were not relevant to the article.

There is no specific paragraph on HSCT, but there is a supplementary table listing several studies in this setting. The authors should explain why they decided not to comment on HSCT studies. 

The table on HSCT has been removed from the article. 

 Line 497:  the authors cited Table 6, but it does not exist

This has been corrected.

Reviewer 3 Report

Comments and Suggestions for Authors

Major comments :

1) Well-written article, but it's too exhaustive a review of the various clinical trials, some of which have been completed, some without encouraging results, for over 5 years! There are already numerous reviews of the various clinical trials, including a very recent one in press on gene therapy ("role of gene therapy in SCD", 2024, in press). We need these updates once or twice a year. However, the authors should distinguish themselves from previous reviews, and no longer mention those that have failed to produce a positive response or had to be stopped early ( IMR -687: Prasugrel ...).

2) For molecules currently available (Hydroxyurea, Voxelotor, Crizanlizumab), we need to talk about ongoing studies, and it's important to point out that Crizalizumab has not yet been granted marketing authorization in Europe: (see ANSM report):  The Phase III study (STAND) in sickle cell disease patients with vaso-occlusive crises did not confirm the clinical benefit of Adakveo. Adakveo was approved in the European Union (EU) in October 2020 for the prevention of recurrent vaso-occlusive crises (VOC) in sickle cell disease in patients aged 16 and over. It could be administered in combination with hydroxyurea/hydroxycarbamide (HU/HC) or as monotherapy in patients for whom HU/HC is inappropriate or inadequate. At the time of its approval in the EU, the data supporting Adakveo's effects were not considered complete due to uncertainty about the magnitude of Adakveo's effect. The drug was therefore granted marketing authorization on condition that the laboratory provided data from the STAND1 study (CSEG101A2301) to confirm the drug's efficacy and safety. The EMA's Committee for Medicinal Products for Human Use (CHMP2) assessed the results of the STAND study and concluded that the clinical benefit of Adakveo was not confirmed.  No new safety issues were identified. However, higher rates of treatment-related grade ≥ 3 adverse events and higher rates of serious adverse events were reported with crizanlizumab compared with placebo. In addition to the STAND study, data from other studies, a compassionate use program and real-life data were reviewed. However, these studies had several limitations, such as a single-arm study design. Consequently, they did not allow us to conclude that there was an effect of Adakveo, and were not sufficient to compensate for the negative results of the STAND study.  In conclusion, as the STAND study did not confirm the clinical benefit of Adakveo, the CHMP concluded in 2023 that its benefit/risk balance is no longer favorable and that its conditional marketing authorization will be revoked in the EU. 

For Voxelotor actually, although there is evidence of voxelotor’s effect on red cell hemolysis in SCD patients, there are scarce data on its effect on other clinical outcomes such as VOC and end organ damage (e.g. brain, kidneys, skin). There is a need for further research in order to understand how voxelotor improves clinical outcomes

3) Authors must more Select Ongoing Trials of Interest in text

Authors should choose between studies that are more fully described in the body of the text and those that are simply cited in the various tables: for greater clarity, and to reduce the length of the tables, only studies not described in the text should be included in the tables. The decision to mention certain studies rather than others should be clearly defined on the basis of objective criteria. 

For example, there's a chapter (3.4) devoted to Etopivat without a word about Mitapivat! In this chapter, we need to talk about ongoing studies on both molecules, with their characteristics: cf. international RISE UP; NCT05031780; EudraCT: 2021-001674-34); Mitapivat needs to be named in Fig.3 if Etopivat is there.

4) some published preliminary results are not reported. For example, preliminary results for Defibrinotide are still published in Blood vol 136, suppl 2020. authors have to talk about it. 

5) Gene therapy chapter  : We need to find a way of classifying the various studies: by their mechanism of action, for example: Gene silencing/Gene editing/Gene correction/Gene addition. In this important chapter, it is difficult to understand why the authors have chosen to talk about certain studies rather than others. 

6) Table 1: why mention the Deferiprone trial (NCT 02443545) on the management of iron overload? It's not a target for the pathophysiology of the disease, but a treatment for a complication. 

7) Table S2: Some recent and ongoing trials in Table S2 could be detailed in the text, as these molecules represent new therapeutic targets: example of crovalimab. Here again, I don't understand the authors' strategies.

On the other hand, in TS2, we find trials on inflammation, metabolism patients in VOC and a PET-scan trial that have no place in a review of therapeutic trials.

Minor comments : 

1) Lig 118: "et al." must be written in italics for et alter throughout the text

2) Table S2: Two studies cited in T. S2 concern marrow transplantation and should be included in T S1 (NCT 01962415 and 03263559).

3) Chap 3.3 RNA-based therapeutics : pathophysiological targets are different Aptamer molecules OX3B and DE3A can be described in chap 2. The aptamer ARC5690 is to be discussed in chap 4 on anti-adhesion molecules.

Author Response

Response to Reviewer 3

Summary

The authors would like to thank you very much for taking the time to review this manuscript. Please find the detailed responses below and the corresponding revisions/corrections highlighted/in track changes in the re-submitted files. Based on your comments, the authors feel that the article is much improved.

Point by point responses

Well-written article, but it's too exhaustive a review of the various clinical trials, some of which have been completed, some without encouraging results, for over 5 years! There are already numerous reviews of the various clinical trials, including a very recent one in press on gene therapy ("role of gene therapy in SCD", 2024, in press). We need these updates once or twice a year. However, the authors should distinguish themselves from previous reviews, and no longer mention those that have failed to produce a positive response or had to be stopped early ( IMR -687: Prasugrel ...).

The authors agree with the reviewer and have removed IMR-687, Prasugrel, Rivipansel, Capsaicin, Sevuparin, and Montelukast, as these drugs have failed to produce a positive response. Additionally, therapies that have been discontinued by the manufacturer have been removed from the tables, including OTQ923 and ARU-1801.

2) For molecules currently available (Hydroxyurea, Voxelotor, Crizanlizumab), we need to talk about ongoing studies, and it's important to point out that Crizalizumab has not yet been granted marketing authorization in Europe: (see ANSM report):  The Phase III study (STAND) in sickle cell disease patients with vaso-occlusive crises did not confirm the clinical benefit of Adakveo. Adakveo was approved in the European Union (EU) in October 2020 for the prevention of recurrent vaso-occlusive crises (VOC) in sickle cell disease in patients aged 16 and over. It could be administered in combination with hydroxyurea/hydroxycarbamide (HU/HC) or as monotherapy in patients for whom HU/HC is inappropriate or inadequate. At the time of its approval in the EU, the data supporting Adakveo's effects were not considered complete due to uncertainty about the magnitude of Adakveo's effect. The drug was therefore granted marketing authorization on condition that the laboratory provided data from the STAND1 study (CSEG101A2301) to confirm the drug's efficacy and safety. The EMA's Committee for Medicinal Products for Human Use (CHMP2) assessed the results of the STAND study and concluded that the clinical benefit of Adakveo was not confirmed.  No new safety issues were identified. However, higher rates of treatment-related grade ≥ 3 adverse events and higher rates of serious adverse events were reported with crizanlizumab compared with placebo. In addition to the STAND study, data from other studies, a compassionate use program and real-life data were reviewed. However, these studies had several limitations, such as a single-arm study design. Consequently, they did not allow us to conclude that there was an effect of Adakveo, and were not sufficient to compensate for the negative results of the STAND study.  In conclusion, as the STAND study did not confirm the clinical benefit of Adakveo, the CHMP concluded in 2023 that its benefit/risk balance is no longer favorable and that its conditional marketing authorization will be revoked in the EU. 

The authors have added information that discusses the failure of the STAND trial to confirm the clinical benefit of crizanlizumab for SCD. Crizanlizumab’s treatment-related adverse effects have also been discussed. Additionally, information regarding crizanlizumab’s conditional approval approval in Europe, and CHMP’s revocation of crizanlizumab’s marketing authorization has been added.

For Voxelotor actually, although there is evidence of voxelotor’s effect on red cell hemolysis in SCD patients, there are scarce data on its effect on other clinical outcomes such as VOC and end organ damage (e.g. brain, kidneys, skin). There is a need for further research in order to understand how voxelotor improves clinical outcomes

The authors agree and have included a section discussing the scarce amount of data regarding voxelotor’s effect on VOC and end-organ damage, and have emphasized the need for further research.

3) Authors must more Select Ongoing Trials of Interest in text

Authors should choose between studies that are more fully described in the body of the text and those that are simply cited in the various tables: for greater clarity, and to reduce the length of the tables, only studies not described in the text should be included in the tables. The decision to mention certain studies rather than others should be clearly defined on the basis of objective criteria. 

The authors have removed clinical trials that have been completed, trials that have limited relevance to the paper, trials that have been mentioned in the text, and trials on drugs that have been discontinued by the manufacturer. Additionally, the authors have decided to add many of the drugs mentioned in the tables to the text. This decision was made based on the existence of currently ongoing clinical trials and preliminary data showing promising results, such as epeleuton and ALXN1820. This has resulted in the removal of the majority of the tables.

For example, there's a chapter (3.4) devoted to Etopivat without a word about Mitapivat! In this chapter, we need to talk about ongoing studies on both molecules, with their characteristics: cf. international RISE UP; NCT05031780; EudraCT: 2021-001674-34); Mitapivat needs to be named in Fig.3 if Etopivat is there.

The authors agree that mitapivat should have been added alongside etavopivat. A section on mitapivat has now been added to the text and Figure 3.

4) some published preliminary results are not reported. For example, preliminary results for Defibrinotide are still published in Blood vol 136, suppl 2020. authors have to talk about it. 

The authors agree and have added preliminary data from Blood vol 136, suppl 2020 to the Defibrotide section.

5) Gene therapy chapter: We need to find a way of classifying the various studies: by their mechanism of action, for example: Gene silencing/Gene editing/Gene correction/Gene addition. In this important chapter, it is difficult to understand why the authors have chosen to talk about certain studies rather than others.

The author would like to thank the review for this suggestion; however, we feel that the recent review on gene therapy classified the various gene therapy drugs based on their mechanism (White, SL 2023).  Thus, we feel that including that would only repeat what was discussed in details in the above article. The objective of this review is to discuss the drugs from the corresponding clinical trials.  Nevertheless, we mention the mechanism of action of the gene therapy drugs in this section.   

6) Table 1: why mention the Deferiprone trial (NCT 02443545) on the management of iron overload? It's not a target for the pathophysiology of the disease, but a treatment for a complication. 

The authors agree and have removed the deferiprone trial from the table.

7) Table S2: Some recent and ongoing trials in Table S2 could be detailed in the text, as these molecules represent new therapeutic targets: example of crovalimab. Here again, I don't understand the authors' strategies.

The authors agree and have added crovalimab to the text along with many other drugs that have ongoing clinical trials and show promise, including ALXN1820, Epeleuton, and Benserazide.

On the other hand, in TS2, we find trials on inflammation, metabolism patients in VOC and a PET-scan trial that have no place in a review of therapeutic trials.

The authors agree and have removed most of the tables, as unrelated clinical trials such as those mentioned by the reviewer have been removed, and most relevant clinical trials in the table have now been mentioned in the text.

Minor comments : 

1) Lig 118: "et al." must be written in italics for et alter throughout the text

The authors have addressed this, and et al. has been italicized throughout the text.

2) Table S2: Two studies cited in T. S2 concern marrow transplantation and should be included in T S1 (NCT 01962415 and 03263559).

The authors have removed this table from the text.

3) Chap 3.3 RNA-based therapeutics : pathophysiological targets are different Aptamer molecules OX3B and DE3A can be described in chap 2. The aptamer ARC5690 is to be discussed in chap 4 on anti-adhesion molecules.

The RNA based therapeutics are now incorporated into the section on “other novel therapeutics”, as the aptamers or nucleic acid therapeutic represent a fairly new class of therapeutic inhibitors.  While the mechanism falls in the other categories, putting it in this section makes more sense. 

Round 2

Reviewer 2 Report

Comments and Suggestions for Authors

In the revised manuscript, the authors fully addressed my comments. 

Author Response

5/10/24

To the Editor:

Below is a point by point response to the reviewers’ comments.  We have tried to address each critique.  We are grateful for the thoughtful review of our work and we feel that it is much improved.

Review Comment Responses

Reviewer 1 comments: - Why have the authors decided to remove the table? –

The authors chose to remove the tables based on the critiques from the other two reviewers.  We included all of the relevant clinical trial numbers in the text.

The authors mentioned that hydroxyurea therapy reduces painful attacks because of the decrease in HbS is INACCURATE; it is because the increase in fetal hemoglobin reduces propensity for sickling.

The authors agree and have removed the mention of HbS reduction from the sentence.

Regarding mitapivat, the authors also do not give credit to the first Phase 1 study for sickle cell disease.

The authors have added information on the 2019 Phase 1 study investigating mitapivat’s safety and efficacy in treating patients with SCD.

And they make voxelotor sound as if it is a wonder drug; it is not because the increase in hemoglobin is illusory; voxelotor does not reduce pain frequency., the tight binding of oxygen impairs off loading of the oxygen.

The authors have toned down the section of voxelotor.  The last paragraph of this section mentions what the reviewer says and we agree with this reviewer’s comment. 

Reviewer 2: In the revised manuscript, the authors fully addressed my comments. 

The authors would like to think the reviewer for the helpful comments. 

Reviewer 3: the authors have shortened and synthesized their review; their message is clearer and more forceful. I agree with their new presentation of gene therapy.
I'll remove Metformin from Figure 3, since the paragraph on this molecule has been deleted.
This article belongs in the up to date reviews of molecules accepted and under evaluation in sickle cell disease. It may be published.

The authors thank you for your time reviewing this paper. The authors have removed Metformin from Figure 3.

Reviewer 3 Report

Comments and Suggestions for Authors

the authors have shortened and synthesized their review; their message is clearer and more forceful. I agree with their new presentation of gene therapy.
I'll remove Metformin from Figure 3, since the paragraph on this molecule has been deleted.
This article belongs in the up to date reviews of molecules accepted and under evaluation in sickle cell disease. It may be published.

Author Response

Reviewer 3: the authors have shortened and synthesized their review; their message is clearer and more forceful. I agree with their new presentation of gene therapy.
I'll remove Metformin from Figure 3, since the paragraph on this molecule has been deleted.
This article belongs in the up to date reviews of molecules accepted and under evaluation in sickle cell disease. It may be published.

The authors thank you for your time reviewing this paper. The authors have removed Metformin from Figure 3.